# The Design of a Low Noise, Multi-Channel Recording System for Use in Implanted Peripheral Nerve Interfaces

**DOI:** 10.3390/s22093450

**Published:** 2022-04-30

**Authors:** Shamin Sadrafshari, Benjamin Metcalfe, Nick Donaldson, Nicolas Granger, Jon Prager, John Taylor

**Affiliations:** 1Department of Electronic and Electrical Engineering, University of Bath, Bath BA2 7AY, UK; bwm23@bath.ac.uk (B.M.); eesjtt@bath.ac.uk (J.T.); 2Department of Medical Physics and Bioengineering, University College London, London WC1E 6BT, UK; n.donaldson@ucl.ac.uk; 3Department of Clinical Science and Services, The Royal Veterinary College, Hawkshead Lane, Brookmans Park, Hatfield AL9 7TA, UK; nicolasgranger@rvc.ac.uk (N.G.); jon.prager@bristol.ac.uk (J.P.)

**Keywords:** neural recording, multiple electrode cuff (MEC), interface filter network, common-mode rejection ratio (CMRR), noise performance, crosstalk, electrode mismatch

## Abstract

In the development of implantable neural interfaces, the recording of signals from the peripheral nerves is a major challenge. Since the interference from outside the body, other biopotentials, and even random noise can be orders of magnitude larger than the neural signals, a filter network to attenuate the noise and interference is necessary. However, these networks may drastically affect the system performance, especially in recording systems with multiple electrode cuffs (MECs), where a higher number of electrodes leads to complicated circuits. This paper introduces formal analyses of the performance of two commonly used filter networks. To achieve a manageable set of design equations, the state equations of the complete system are simplified. The derived equations help the designer in the task of creating an interface network for specific applications. The noise, crosstalk and common-mode rejection ratio (CMRR) of the recording system are computed as a function of electrode impedance, filter component values and amplifier specifications. The effect of electrode mismatches as an inherent part of any multi-electrode system is also discussed, using measured data taken from a MEC implanted in a *sheep*. The accuracy of these analyses is then verified by simulations of the complete system. The results indicate good agreement between analytic equations and simulations. This work highlights the critical importance of understanding the effect of interface circuits on the performance of neural recording systems.

## 1. Introduction

### 1.1. Neural Recording

The recording of signals from the peripheral nervous system (PNS) (the *electroneurogram*—ENG) using chronically implanted electrodes is one of the major challenges in current neuroprosthetic research. Several types of implantable interfaces have been proposed, but very few have been validated with long-term chronic studies. One of the most well-established types, both for stimulation and recording, is the extraneural nerve *cuff* [1]. Cuffs are widely used for the electrical stimulation of the peripheral nerves, including in commercial devices, such as the LivaNova vagus nerve stimulator for intractable epilepsy [2,3]. Likewise, there have been several demonstrations of recording from the PNS using cuffs, albeit predominantly in *acute* studies [4]. *Tripolar* stimulation cuffs, in which a current is driven between the centre electrode and the outer pair of electrodes (which are usually connected together) are also frequently employed [5]. For recording purposes, a monopole, dipole or tripole electrode structure is connected to a differential amplifier or a double-differential amplifier. The dipole recording configuration is most common, although it has been shown that the tripole arrangement can reduce interference by improving common-mode (CM) rejection [6]. 

The neural signals within the PNS may be classified as *afferent* or *efferent*, corresponding to the direction of propagation. Most peripheral nerves (especially when interfaced using an extraneural approach) are *mixed* and contain many afferent and efferent axons. In order to record selectively from specific axons, it is necessary to perform some form of signal processing to separate the propagating signals. There are many approaches to this problem, such as the use of spatio-temporal filters [7], source localisation [8], electrical impedance tomography [9], or discrimination based on conduction velocity [10]. *Velocity Selective Recording* (VSR) uses multi-electrode cuffs (MECs) to detect and classify neural signals based on the different velocities present, exploiting the correspondence between conduction velocity and fibre diameter, at least for myelinated nerves [11,12]. 

### 1.2. Recording Challenges of MECs

To implement VSR, the electrodes of the MEC must be connected to a bank of differential amplifiers [11] and this increases the complexity of the amplifier design task. As for the dipoles and tripoles, the amplitudes of *spontaneous* (i.e., naturally occurring) neural signals recorded are very small (generally less than 1 µV [13]). Therefore, the differential voltage gain must be high (typically 60–100 dB) with an adequate *signal-to-noise ratio* (SNR), requiring very low noise front-end amplifiers, i.e., the noise floor of the amplifier must be less than a few nV/√Hz [14]. The outline schematic of an example recording system is shown in Figure 1, where *N* (typically about 10) electrodes are fitted to an insulating nerve cuff forming a MEC. The electrodes are shown connected directly to an amplifier array as *dipoles*. 

For the MEC, in addition to the exacting differential specification required of each amplifier, in order to mitigate the effects of undesired interfering signal sources, CM gain and crosstalk between the individual channels must be minimised. In the recording of spontaneous neural signals, interference comes from nearby muscles, AC mains and radio frequency (RF) pick-up [15]. Singly or in combination, such interference can saturate high-gain amplifiers. Therefore, a bandpass filter interface network, placed between the recording electrodes and the front-end amplifiers, is essential to limit the effects of high- and low-frequency interfering signals. Unfortunately, the presence of such a filter network conflicts with the noise and CM rejection behaviour of the amplifiers and so the interface between the two requires careful design. Moreover, the impedance mismatch between channels in a multi-electrode recording system is inevitable and will also affect the system’s performance. In this paper, two interface circuits suitable for a multi-electrode recording system are examined for cuff electrodes. Analyses of noise, *common-mode rejection ratio* (CMRR) and crosstalk are presented for two input networks (‘Type 1’ and ‘Type 2’) (these correspond to the networks called ‘Type 1’ and ‘Type 2’ in [15]). The two networks were originally introduced and analysed for the case of a single tripole (one differential amplifier) in [16], and it was found that the ‘Type 2’ arrangement was superior to the ‘Type 1’ network in terms of both CMRR and noise performance. In this paper, we extend and enlarge the results of [16] and demonstrate that the single-amplifier results do *not* extend to a multi-electrode system. The two networks are shown in Figure 2.

The overall intention of this work is to aid and inform the designer of such a system and provide the theory that simulation alone cannot. Having said that, the validation study presented is based mostly on simulation. The justification for this approach is discussed in Section 4 and relates to the inherently low sensitivity of the circuits employed to component tolerances (including the electrode impedances) and the reliability of simulation models in the bandwidth under consideration. Equations are derived symbolically using MATLAB and simplified for practical component values, reflecting how the various parameters affect the design criteria. Although the equations are derived for a 10-channel recording system, they can be extended and generalised for an *N* electrode system. SPICE simulations and some measured data are used to validate the accuracy of the procedure and indicate good agreement between the detailed analytic equations and the simplified versions. 

The outline of this paper is as follows. CMRR, crosstalk and noise performance of a 10-channel multi-electrode system with two types of filters are discussed, and corresponding approximate equations are derived in Section 2. Section 4 investigates the accuracy of the approximate equations derived in Section 2 using SPICE simulation. Section 5 discusses how these analyses can help designers to achieve the desired specifications in a multi-electrode recording system and, finally, conclusions are drawn in Section 6.

## 2. Interface Circuits

Figure 2a,b show the two candidate front-end circuits that are referred to as ‘Type 1’ and ‘Type 2’ [16]. Both circuits place an *RC* bandpass filter between the cuff electrodes and the input terminals of the amplifiers, realised by first-order high pass and low pass sections connected in cascade, the time constants being realised by the series and parallel capacitors (*C_s_*, *C_p_*), respectively, and associated resistances, including the two impedances *R_d_* and *R_e,_* (see below). The cut-off frequencies are chosen to be about 100 Hz and 100 kHz [14], respectively, attenuating both low- and high-frequency interfering signals while satisfying the bandwidth requirements of a 10-channel VSR system for neural recording [7]. Note, also, that the sensitivities of the circuit transfer functions to the various component values (including those associated with the electrodes) will be *low*.

In all cases, the electrodes, which are grouped as *dipoles*, are represented by an equivalent circuit consisting of a voltage source (*V_d_*) in series with an impedance (*R_d_*) representing the axial component of the section of the tissue inside the cuff. This combination of voltage generator and impedance appears to be a generic Thévenin source, but actually, it is more complicated than that. The source will be the superposition of voltages from all the separate axons, and each axon produces action potentials at the dipole electrodes which is the product of *R_d_* and its action current—the action current is being determined by the travelling transmembrane (TMAP) potential and the high axon resistance [11]. The action currents are therefore a property of the nerve and only *R_d_* can be altered by the design of the cuff. It follows that *V_d_* and *R_d_* are not independent but, in fact, *V_d_* is proportional to *R_d_*. *R_d_* is resistive and has a typical value of about 0.5 kΩ [9]. By contrast, the impedances of the electrodes connecting the tissue to the front-end amplifier circuits (*R_e_*) are complex, with typical values (moduli) of about 1 kΩ at a frequency of 1 kHz. These element values and assumptions have been employed in assessing the performances of the two candidate circuits discussed in the paper and to simplify the design equations.

The only difference between the two circuits in Figure 2 is that in the Type 1 circuit of Figure 2a the bias current path to ground for each amplifier input is provided by individual resistors (*R_a_*) while in the Type 2 configuration of Figure 2b, a centre tapped pair of resistors (*R*_1_) is employed [8]. The two resistors’ *R_cm_* represents the impedances between the reference electrode (normally placed far away from the cuff) and the end electrodes of the cuff. *V_cm_* is the CM input voltage (i.e., the interfering source in this case). At *passband* (mid-band) frequencies, it is assumed that the shunt capacitors become open circuits and the series capacitors short circuit. In the following sub-sections, expressions for CMRR, crosstalk and noise are derived for both versions of the circuit, for passband operation in all cases. 

Finally, it is worth noting that all the circuits employed are based on simple first-order *RC* circuits and, since no overall feedback is employed, the poles of the transfer functions lie on the negative real axis of the complex plane. This has the advantage that *Q* values are low in the frequency domain and hence sensitivity to component tolerances is also low.

### 2.1. Common-Mode Rejection Ratio (CMRR)

Using an amplifier with high CMRR does not guarantee that the entire system will operate with high CMRR. A front-end filter network of the type shown in Figure 2 (either version) can generate differential signals from common-mode inputs, reducing the overall CMRR of the system. It can be shown that the CMRR of the entire system is related to the CMRR of the individual amplifiers by the following expression [8]: (1)$$\frac{1}{\mathrm{CMRR}}\approx \frac{1}{{\mathrm{CMRR}}_{a}}+\frac{V_{ai}}{V_{cm}}$$
where ${\mathrm{CMRR}}_{a}$ is the amplifier’s common-mode rejection ratio and *V_ai_/V_cm_* represents the differential gain from the common-mode input (see Figure 2—both forms) to the *i*′th differential input of the amplifier array. Therefore, an inappropriately designed front-end filter can degrade the performance of the entire system, irrespective of the quality of the amplifiers. To investigate the effect of the front-end network on the overall CMRR, the common-mode gain at the differential input of each amplifier *V_ai_* was calculated for both candidate structures, for *N* = 10.

#### 2.1.1. ‘Type 1’ Biasing Structure

To calculate the effect of a common-mode signal at the inputs of the amplifiers, all differential input voltage sources *Vd_i_* are set to zero, and the KCL equations for nodes *V*_1_, …, *V*_10_, *V′*_1_, …, *V′*_10_ are written in the form:(2)$$A\times \left[\begin{array}{c} V_{1}\\ :\\ \begin{array}{c} V_{10}\\ {V^{\prime }}_{1}\\ :\\ {V^{\prime }}_{10} \end{array} \end{array}\right]=B\times V_{cm}$$
where **A** is the (20 × 20) connection matrix shown in Table 1 and **B** is a (20 × 1) column vector. In **B**, all elements are zero except for the 11th and 20th, which are *G_cm_*_1_ and *G_cm_*_2_, respectively.
(3)$$B=\left[\begin{array}{c} \begin{array}{c} 0\\ 0\\ : \end{array}\\ \begin{array}{c} 0\\ G_{\mathrm{cm}1}\\ 0 \end{array}\\ \begin{array}{c} \begin{array}{c} 0\\ : \end{array}\\ G_{\mathrm{cm}2} \end{array} \end{array}\right]$$

In the analysis, the conductances *G_cm_*, *G_d_*, etc., are the inverses of the resistances *R_cm_*, *R_d_* in Figure 2. Assuming that the amplifiers are ideal, *V*_1_, …, *V*_10_, *V′*_1_, …, *V′*_10_ can be calculated as:(4)$$\left[\begin{array}{c} V_{1}\\ :\\ \begin{array}{c} V_{10}\\ {V^{\prime }}_{1}\\ :\\ {V^{\prime }}_{10} \end{array} \end{array}\right]=A^{-1}\times B\times V_{cm}$$

The nodal equations for *V_1_–V_10_* were solved symbolically using MATLAB (see Appendix A for an example equation). The input impedance (*R_ai_*) should be larger than the electrode impedances in order to maximise the voltage drop across the input. Therefore, making the reasonable assumptions that:*R_ai_* >> *R_ei_*;*R_ai_* >> *R_cmi_* and*R_ai_* >> *R_di_*; all *i*
and all the *R_a_* are equal, all the *R_e_* are equal, and the *R_cm_* are equal, the additional CM gain in the *i*’th channel due to the filter network can be expressed as follows:(5)$$\frac{V_{ai}}{V_{cm}}\approx \frac{\left(\frac{N}{2}-i\right)R_{d}}{R_{a}}\;;i=1,2,\;\ldots ,N/2$$
where *V_ai_* is the differential input voltage to the *i*’th amplifier and *N*, the number of electrodes is assumed to be even (similar arguments apply for *N* odd) and with a value of 10 in this case. Equation (5) shows that the outer channels have higher common-mode gain and therefore lower CMRR than those near the centre of the array, where the overall CMRR approaches that of the amplifiers. Note that the additional CM gain can be reduced by reducing the ratio *R_d_*/*R_a_* as far as possible. 

#### 2.1.2. ‘Type 2’ Biasing Structure

Figure 2b shows the front-end network employing the ‘Type 2’ biasing circuit. As in the case of the ‘Type 1’ structure, KCL equations can be derived and the voltages *V*_1_–*V*_10_ calculated. Proceeding as in the ‘Type 1’ case, assuming this time that for all *i*, *R*_2*i*_ >> *R_cmi_, R_di_* and *R_ei_* and that, in addition, *R_ei_* << *R*_1*i*_, the CM gain at the differential input of the amplifier is approximated as:(6)$$\frac{V_{ai}}{V_{cm}}\approx \frac{\left(\frac{N}{2}-i\right)(R_{d}||2R_{1})}{R_{2}};\;i=1,\;2,\ldots ,N/2$$

Equation (6) shows that increasing *R*_2_ and reducing *R*_1_ improves the CMRR. However, note that reducing *R*_1_ also decreases the differential gain of the channels in addition to possibly reducing the accuracy of the approximation, which requires, as already noted, that *R_ei_* << *R*_1*i*_. Therefore, some design trade-offs may be required in this case.

### 2.2. Crosstalk between the Channels

The differential gain from each input source *V_di_* to the amplifier inputs *V_aj_* can be discussed in two parts. Firstly, the gain from each source to its corresponding amplifier input should be as close to unity as possible and, secondly, the gain from each source to the inputs of the other amplifiers, which, for the purposes of this paper, we refer to as *crosstalk*, should ideally be zero. In this section, the differential gain at each amplifier input is calculated for both candidate structures.

#### 2.2.1. ‘Type 1’ Biasing Structure

To analyse the differential gain at each amplifier input *V_aj_*, all the differential inputs *V_di_* except one are set to zero (the CM input voltage is also set to zero). For this calculation, it is convenient to use the *Norton* form of the equivalent circuit for each source, where:(7)$$I_{di}=\frac{V_{di}}{R_{di}}$$

The KCL equations for *V*_1_, …, *V*_10_, *V′*_1_, …, *V′*_10_ are as follows:(8)$$A\times \left[\begin{array}{c} V_{1}\\ :\\ \begin{array}{c} V_{10}\\ {V^{\prime }}_{1}\\ :\\ {V^{\prime }}_{10} \end{array} \end{array}\right]=C_{i}\times I_{di}\;$$
where ***A*** is defined in Table 1 and the column vector ***C_i_*** is: (9)$$\;C_{i}=\left[\begin{array}{c} c_{1}\\ :\\ c_{k}\\ :\\ c_{10} \end{array}\right]\;\;\;,\;c_{k}=\{\begin{array}{cc} 1 & k=i\\ -1 & k=i+1\\ 0 & else \end{array}\;$$

The voltages *V*_1_–*V*_10_ follow from:(10)$$\left[\begin{array}{c} V_{1}\\ :\\ V_{10} \end{array}\right]=A^{-1}\times C_{i}\times I_{di}\;$$
and the differential gain from each source to each amplifier input is given by Equation (11):(11)$$\frac{V_{aj}}{V_{di}}=\{\begin{array}{cc} 1-\frac{R_{d}}{2R_{cm}+9R_{d}} & if\;i=j\\ \frac{-R_{d}}{2R_{cm}+9R_{d}} & if\;i\neq j \end{array}\;$$

Equation (11) shows that crosstalk is reduced by reducing the ratio *R_d_/R_cm_*.

#### 2.2.2. ‘Type 2’ Biasing Structure

The analysis is very similar to that employed for the ‘Type 1’ structure and the differential gain in this case is:(12)$$\frac{V_{aj}}{V_{di}}=\{\begin{array}{cc} \frac{R_{d}||2R_{1}}{R_{d}}\left(1-\frac{R_{d}||2R_{1}}{2R_{cm}+9(R_{d}||2R_{1})}\right) & if\;i=j\\ -\frac{R_{d}||2R_{1}}{R_{d}}\left(\frac{R_{d}||2R_{1}}{2R_{cm}+9(R_{d}||2R_{1})}\right) & if\;i\neq j \end{array}\;$$

As in the case of the ‘Type 1’ structure, crosstalk in the ‘Type 2’ biasing arrangement is reduced by decreasing the ratio of *R_d_/R_cm_*. Note, in addition, that in this case, decreasing *R*_1_ reduces both crosstalk and differential gain, which may be unacceptable in some applications.

### 2.3. Noise Analysis

In this section, the main sources of noise and their effect on the behaviour of the circuit, including the front-end filter networks (as shown in Figure 2) are described. The subject of noise in multi-channel VSR systems was dealt with comprehensively in [12]. In this paper, we present a simplified approach based on the earlier work, however, we intend to inform the designer of the main issues involved and their possible mitigation. The contributions of *thermal* noise, *amplifier voltage* noise and *amplifier current* noise are considered separately, and the individual sources are assumed to be *Gaussian* and *uncorrelated* in all cases. 

In both the Type 1 and Type 2 arrangements, the axial components of the electrode/tissue combination are denoted *R_d_*, while the *R_e_* is the electrode impedances (see Figure 2). The two resistances *R_cm_* are the common-mode elements connecting the ends of the cuff system to the reference (shown as ground in the figure). In the Type 1 circuit, the *R_a_* are bias resistors, essentially used to define the DC bias point of the amplifier inputs, whereas, in the Type 2 circuit, a balanced-*T* network is used for this purpose. As already noted, one consequence of this arrangement is that a potential divider exists involving *R_e_* and *R*_1_, that influences both the signal and noise amplitudes. 

#### 2.3.1. Thermal Noise

For noise modelling, as in other sections of the paper, all the resistors are assumed to be *real* (i.e., ohmic) except for *R_e_*, which are complex impedances. In the passband of the filter, the capacitors are chosen, so that those connected in series are short circuits while those in shunt are open circuits. The following assumptions amongst the impedances are made:

*R_ai_ >>**∣**R_ei_**∣*;


*R_ai_ >> R_cmi_,*


*R*_2*i*_ >> *R*_1_*_i_* and

*R_ai_ >> R_di_*; all *i*

For noise calculations, it is helpful to redraw the filter circuit in the passband as a ladder network, where the impedances of each type are assumed to be equal. For the Type 1 circuit, it is as shown in Figure 3a. 

Noting the inequalities above, the *R_a_* appear in parallel with much smaller resistances and so can be omitted from the noise calculations. The simplified circuit is redrawn, as shown in Figure 3b. 

If each *R_d_* is removed from the circuit in turn, the Thévenin equivalent circuit of residual impedance at the port where the resistor was removed is [(*N* − 1)*R_d_* + 2*R_cm_*]. For *N* = 10 (i.e., large) and, given that R_cm_ is not negligible, this residue will be in excess of 10*R_d_*. This resistance appears in parallel with the *R_d_* removed and so it is reasonable to assume that the only significant contribution to *thermal* noise density at the input to an amplifier is given by one *R_d_* and the associated pair of impedances *R_e_*. The thermal noise density contribution appearing at the input to this amplifier, expressed as an rms voltage, is, therefore:(13)$$V_{a1}\left(rms\right)=\sqrt{4kT\left(R_{d}+2re(R_{e}\right)}$$
where *re*(*R_e_*) represents the real part of *R_e_.*

Similar expansion and simplification procedures can be applied to the Type 2 circuit, resulting in the equivalent circuit shown in Figure 4. As already noted in Section 2.1.2 of the paper, the presence of the resistors *R*_1_ places a potential divider (involving the impedances *R_e_* as the other element) in each signal path. Due to the small size of the recorded signals, it is highly desirable to make the gain of this divider as close to unity as possible and so we can add a further inequality to the previous list:

*R*_1*i*_ >> ∣*R_ei_*∣, all *i*

**Figure 4 sensors-22-03450-f004:**
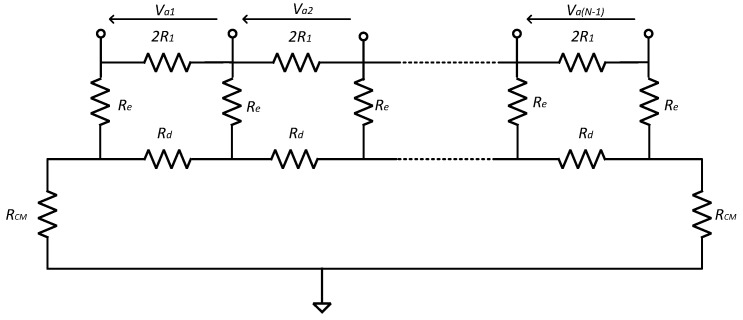
Simplified passband equivalent circuit for thermal noise analysis for a Type 2 circuit. This is a ladder expansion of Figure 2b, to which the same simplifying procedures have been applied as were used in Figure 3a. In addition, as already noted, for maximum signal gain we require that *R*_1_ >> ∣*R_e_*∣ and so the circuit reduces to that of Figure 3b.

A consequence of this is that the *R*_1_ will contribute little to the overall noise calculation since they also appear in parallel with many small resistances. The thermal noise equivalent circuits for both Type 1 and Type 2 circuits are therefore the same (Figure 3b) and the thermal noise density appearing at the input to each amplifier is given by Equation (1).

#### 2.3.2. Amplifier Noise

Figure 5 is the noise equivalent circuit of a single-ended output, differential voltage amplifier, where *v_n_* and in are rms noise sources (densities), respectively. Assuming the input resistance of the amplifiers is so large as to be effectively infinite, the voltage noise contribution of each amplifier appears only at the corresponding output, since no current flows into the other parts of the circuit as a result of the presence of *v_n_*.

The equivalent circuit for the amplifier *current noise* is shown in Figure 6, where the simplified form of the circuit developed above has been used. It is clear from this circuit that each amplifier will cause noise current to flow around a primary loop consisting of one *R_d_* and two *R_e_* and also around a secondary loop consisting of (*N* − 1) resistors *R_d_* and two *R_cm_*. Using the same arguments as for thermal noise, the current in the secondary loop will be significantly less than in the primary loop and can be ignored for practical purposes. Using the principle of superposition (i.e., taking each source individually, the others being removed from the circuit) and treating the various resistances and impedances as noiseless (since the noise contributions of these components have been considered separately), the contribution to the rms voltage *V_a_*_1_ due to *i_a_*_1_ is:*V_a_*_11_ = *i_a_*_1_|*R_d_*_1_ + *R_e_*_1_ + *R_e_*_2_|

Since *V_a_*_1_ appears across one of the pairs of electrodes at the end of the array, considering only currents flowing in the primary loops, the only other contribution comes from *i_a_*_2_: *V_a_*_12_ = *i_a_*_2_|*R_e_*_2_|

Assuming that all the *R_d_* are equal and all the *R_e_* are equal and that *i*_*a*1_ and *i*_*a*2_ are uncorrelated sources of the same statistics and of equal amplitude *i_a_*, the total rms value of *V*_*a*1_ is given by summing the two contributions, recalling that *R_e_* is a complex impedance:$$V_{an}=i_{a}\sqrt{{\left|R_{d}+2R_{e}\right|}^{2}+{\left|R_{e}\right|}^{2}}$$
where *n* = 1 or *N* − 1, and, similarly, for the other inputs (i.e., those *not* placed at the ends of the array):$$V_{am}=i_{a}\sqrt{{\left|R_{d}+2R_{e}\right|}^{2}+2{\left|R_{e}\right|}^{2}}$$

In this section, in summary, expressions have been derived for the thermal noise and amplifier voltage and current noise appearing at the input of one of the amplifiers in the *N* channel system shown in Figure 2. The analysis is a simplified form of that given previously in [12] and, given certain assumptions, ensures the accuracy of the thermal and amplifier current noise calculations is better than approximately 1/*N*, where *N* is typically 10 (the amplifier voltage noise calculation is exact). Since all three sources are assumed to be white and uncorrelated, an expression for the total input-referred noise density for one amplifier can be written using superposition. This is valid for both Type 1 and Type 2 arrangements: (14)$$v_{a\left(\mathrm{total}\right)}=\sqrt{4kT\left[R_{d}+2re\left(R_{e}\right)\right]+v_{n}^{2}+i_{a}{}^{2}({\left|R_{d}+2R_{e}\right|}^{2}+p{\left|R_{e}\right|}^{2})}$$
where *p* = 1 at the end amplifiers, and *p* = 2 for the others.

Table 2 summarizes the equations derived in this section. 

## 3. Electrode Impedance Mismatch

In Section 2, the calculation of CMRR, crosstalk and noise performance for the case of matched electrode impedances was discussed. However, in practice, impedance variation of the individual electrodes, as well as the mismatch between the electrodes, is inevitable. Typical reasons for this are fabrication process non-idealities and inconsistencies and the growth of encapsulation tissue around the electrodes. In addition, changes in electrode size and separation would affect the impedances of the electrodes (*R_d_* and *R_e_*), and hence, both the gain and the upper cut-off frequency of the bandpass filter to some extent. However, the inherently low sensitivity design of the filter circuits ensures that the effect on the system parameters would be small.

As an illustration of typical impedance variations, a 10 electrode MEC with stainless steel ring electrodes (diameter 1 mm, electrode pitch 1.5 mm) was fabricated and implanted on the second sacral spinal nerve root S2 (left) of a female sheep. Impedance measurements were taken using a two-wire configuration at 1 kHz with a 100 mV source voltage. The measurements are shown in Table 3 and indicate that the modulus of the impedance mismatch between pairs of electrodes may be as high as 200%, and therefore *not* negligible (note that the impedance values in Table 3 are illustrative *only*. This is because these values include contributions from both *R_e_* and *R_d_*. At present we do not have estimates of these impedances separately). Clearly, then, it is critically important that the impedance mismatch is included in the design and analysis of the recording instrumentation.

To illustrate the effect of impedance mismatch, the effect on CMRR and crosstalk are discussed for both types of interfaces. As in the case of the matched interface networks, analysed in Section 2, KCL equations are derived, and the CM gain and crosstalk are calculated for both interfaces with mismatched electrode impedances. Note, also, that the same assumptions are applied as in Section 2. As a result, only the axial component of the electrode impedance (*R_d_*) appears in the equations in Table 4 and Table 5 and only variations of *R_d_* and *R_cm_* need to be considered. 

The effect of mismatch between two reference impedances (*R_cm_*) on the gain is shown in Table 4, where *R_cm_* and Δ*R_cm_* represent the average value of the reference impedance and its mismatch. For both circuits, the first term of the CM gain equations is the same as the gain of a network without mismatch. The second term is added to reflect the *R_cm_* mismatch and therefore degrades the CMRR. However, it does *not* affect the differential gain and so crosstalk is not affected by *R_cm_* mismatch. 

The effect of axial impedance mismatch on common-mode and differential gain is summarised in Table 4. In these equations, *R_di_* represents the axial impedance of the *ith* electrode. Therefore, the mismatch of each electrode affects the CMRR and crosstalk of its corresponding channel *only* in both types of structure.

## 4. Validation by Simulation

### 4.1. Accuracy of the Approximate Equations

To verify the accuracy of the approximate equations derived in Section 2, a 10-channel interface was simulated (using SPICE) for both the ‘Type 1’ and ‘Type 2’ structures. The amplifier is assumed to have a high input impedance and not to interact with the interface, except for the amplifier’s current noise source. As already noted, this causes currents to flow in the interface circuit, resulting in differential voltages at the amplifier inputs. The effect of the amplifier current noise source is therefore expressed as an *impedance*. The other simulations considered are CMRR, crosstalk and thermal noise (note that the input-referred *rms* voltage noise does not depend on the design of the interface circuit and so is not considered). The comparison between the two sets of results for CMRR is shown in Figure 7 for different cases where some parameters are kept constant while others are swept over a practical range of interest (note that for compactness, only the curves describing the CMRR performance are shown). The variables chosen for this validation process are as shown in Table 6.

In Figure 7, the *x*-axes show the swept parameters, the others being held at their nominal values. The CMRR variation as a function of *R_d_* and *R_a_* (‘Type 1’)/*R*_2_ (‘Type 2’) is shown in Figure 7a,b. The inaccuracies caused by the various approximations are less than 0.05 dB compared with the SPICE simulations. Similarly, for *crosstalk*, where *R_d_* and *R_cm_* are varied, the inaccuracies caused by the approximations are less than 0.1 dB. The errors in the calculation of thermal noise and amplifier current noise gain are less than 3.6% and 0.08%, respectively.

### 4.2. Simulation of the Complete System

As an example of the characterisation of a complete recording system, a 10-channel ‘Type 1’ interface network (Figure 2a operating in the passband), connected to a low noise CMOS amplifier array, was simulated using SPICE. In this example, all the components in the interface network were *matched* (*R_a_* = 10 MΩ, *R_cm_* = *R_d_* = 1 kΩ). The specifications of the ENG amplifier are summarised in Table 7.

The simulated and calculated results are shown in Table 8. It should be noted that for the calculated CMRR value, the measured amplifier CMRR (77.5 dB) is included (Equation (1)), whereas neglecting the effect of the amplifier results in a CMRR of 67.96 dB. For the noise density, Equation (14) calculates the total noise density referred to as the input of the amplifier, which results in $8.5\;\mathrm{nV}/\sqrt{\mathrm{Hz}}$ and, in order to refer the noise to the voltage source *V_di_*, it should be divided by the corresponding gain from (Equations (11) and (12)). For both crosstalk and noise parameters, the calculated values and simulation results are in good agreement, as illustrated in Table 8. Unfortunately, as explained above, it is not possible at present to incorporate the measured results given in Table 4 into these simulations, as separate values for *R_d_* and *R_e_* are not available.

## 5. Discussion

### 5.1. Validity of Assumptions

#### 5.1.1. Use of a Simulation-Based Analytical Study

As noted above, with few exceptions, the study presented in this paper is based on *simulation*. The justification for this is that: (a) the circuits employed to achieve a bandpass characteristic (first-order *RC* combinations) have inherently low sensitivity to parameter values; (b) that the bandwidth (centred around 1 kHz) is very low; (c) the circuits are *fee-forward*, i.e., no overall feedback is applied. Taken together, these aspects favour the use of simulation because (i) the use of *RC* circuits for frequency selection and the absence of overall feedback means that the transfer functions of the circuits have low *Q* in the frequency domain and, therefore, have low sensitivity to component tolerance. Furthermore, (ii) in the bandwidth of the systems discussed, the component models employed in commercial circuit simulators provide an adequate level of precision for most practical purposes. 

#### 5.1.2. The Use of Gaussian Noise Models

This paper has presented a detailed analysis of the noise performance (in terms of thermal noise, amplifier voltage noise and amplifier current noise) of the Type 1 and Type 2 circuits, with a focus on the design guidelines that aim to maximise the SNR. Throughout this analysis, the noise sources have been modelled using independent Gaussian processes—a common approach for noise modelling. Interference has been assumed to arise predominantly from high-frequency radio transmissions (RF) and would be dealt with via the appropriate selection of the corner frequencies of the bandpass filter. This is made possible because the RF sources are significantly out-of-band when compared to ENG. However, other biological sources within the body will also contribute to the interference, and it is likely that some of these will produce impulsive noise that may be better represented by coloured noise (e.g., Brownian noise). These sources will most likely be in-band, and thus *not* readily removed by simple filtering in the frequency domain. These sources are not considered in this paper; however, they would most likely appear at the recording interface as correlated sources akin to the ECG artefact. Thus, the analysis of the common-mode rejection ratio considered in Section 2.1 would apply in this case. 

### 5.2. Optimisation of Component Values

A set of analytical design equations were derived in Section 2, linking the main system parameters (CMRR, crosstalk and noise) for two types (Types 1 and 2) of practical multi-channel interface networks to the underlying component values. These equations were validated (by simulation) in Section 3 and found to be accurate in all cases, at least when reasonable assumptions were made. In fact, it was noted that if *R*_1_ >> *R_d_*, and *R*_2_ ~ *R_a_,* the same design equations can be used for both types of interface networks, suggesting a unified graphical presentation to aid the designer. This is important, since the design of a multi-channel interface system, optimised to a particular specification, presents the designer with a very large set of design choices and possibilities. 

The four plots in Figure 8 show the design equations for CMRR, crosstalk, thermal noise and amplifier current noise derived in Section 2 for both interface structures in a compact, visually accessible form. Figure 8a (based on Equation (5)) shows that CMRR is increased by increasing the values of the grounded resistors (*R_a_*) and/or reducing the cuff impedances (*R_d_*), always recognising that the upper limit of CMRR is determined by the amplifier itself. Note that crosstalk (Figure 8b and Equation (11)) can be reduced by reducing the ratio of *R_d_* to *R_cm_*, but for practical values of these elements, crosstalk is fairly constant. 

In Figure 8c,d, the effects of thermal noise and gain of amplifier current noise at preamplifier input are plotted at constant temperature (37 °C) in terms of *R_d_* and *R_cm_*, assuming that Re is negligible in comparison with other impedance values. These plots demonstrate the effect of the interface network impedances on the components of the total input-referred noise that depends on them, i.e., neglecting the voltage noise contribution of the amplifier. Both these parameters can be reduced by decreasing the electrode impedance (*R_d_*) but are both almost independent of the common-mode resistance, *R_cm_*. Note, however, that since the total input-referred noise is the superposition of all three components (see Equation (14)), reducing any or all of them will reduce the total noise. 

In summary, ∣*R_e_*∣ should be small because this will reduce both thermal and amplifier current noise (Equations (13) and (14)) and avoid attenuation of the neural signal. Equations (13) and (14) are valid for the noise, however, they need to be considered carefully when optimising the design. The noise is minimised by reducing both *R_e_* and *R_d_*, however, because the signal amplitude is proportional to *R_d_* [11,17], in practice, to maximise signal/noise, the cuff should be designed to maximise *R_d_* and the effect that has on noise is unavoidable. *R_cm_* should be as large as possible in order to minimise crosstalk (Equations (11) and (12)), which will only have a small effect on noise. Increasing *R_a_* (Type 1) or *R*_2_ (Type 2) reduces CMRR (Equations (5) and (6)); the analysis in Section 3 does not show any difference between the two biasing networks. Finally, note that the maximum value of *R_a_* is limited by the input impedance of the amplifier (possible potential divider effects reducing the signal amplitude) and the effect on input filter design (time constants, etc.).

The analysis in this paper is for networks operated in the passband. However, for a complete filter design, the capacitor sizes should be determined based on the resistor values chosen using the criteria discussed in this paper. The required lower and upper cut-off frequencies determine the *C_s_* and *C_p_* capacitor sizes, respectively. It should be noted that to have equal cut-off frequencies, all the *C_s_* should be equal, and all the *C_p_* should be equal as well.

## 6. Conclusions

Bandpass filter networks are essential for interfacing arrays of electrodes (such as nerve cuff electrodes) to amplifier arrays for applications, such as *velocity selective recording* (VSR). However, the inclusion of these networks influences the CMRR, crosstalk, and noise performance of such systems. In this paper, an analysis has been provided relating the performance criteria of two commonly used front-end biasing networks to component values. The effect of electrode mismatches is also discussed for both networks. It is shown that the ‘Type 1’ and ‘Type 2’ biasing networks result in a similar performance, provided that certain assumptions are satisfied. 

The approach taken in the paper is to generalise the design problem as far as possible in order to aid the designer in the complex task of designing an interface network for a particular application. Using reasonable assumptions, the complicated state equations of the complete system are simplified into a manageable set of design equations, which are then verified for accuracy by comparison with the SPICE simulations of the complete system. Finally, a set of three-dimensional plots has been derived, which show how the main parameters of interest (CMRR, crosstalk and noise) depend in a very general way on the circuit component values. The knowledge thus provided to the designer will help in the construction of a first, probably approximate design, that can subsequently be refined by simulation. 

## Figures and Tables

**Figure 1 sensors-22-03450-f001:**
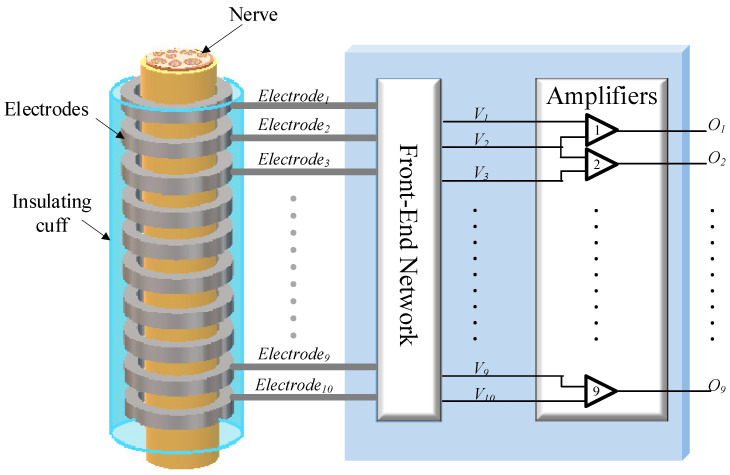
Simplified schematic of a typical recording configuration using a MEC fitted to a peripheral nerve. There may be one or more ranks of amplification after the front-end network.

**Figure 2 sensors-22-03450-f002:**
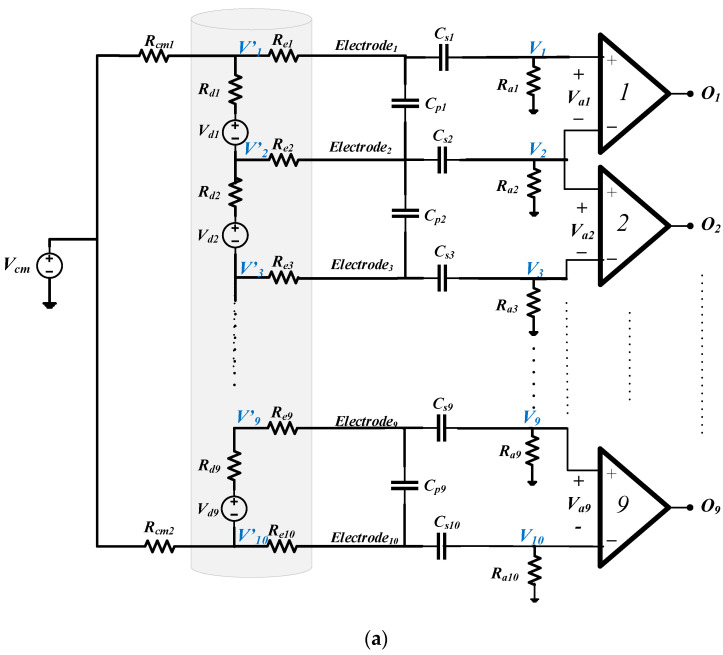
Front-end network with (**a**) ‘Type 1’ biasing arrangement and (**b**) ‘Type 2’ biasing arrangement.

**Figure 3 sensors-22-03450-f003:**
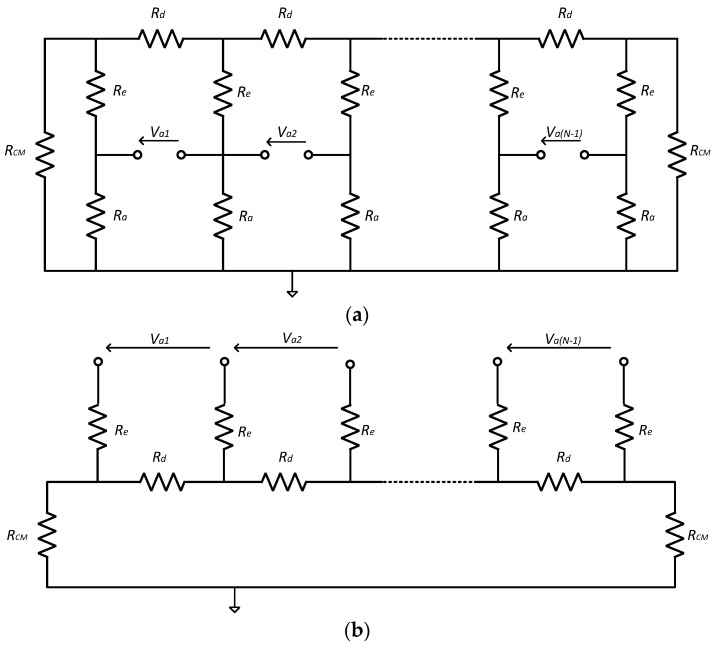
(**a**) Passband equivalent circuit for thermal noise analysis. This is a ladder expansion of the circuit of Figure 2a, i.e., for a Type 1 circuit. (**b**) The circuit of (**a**) with the resistors *R_a_ is* removed to reflect the fact that they appear in parallel with much smaller resistors and therefore contribute little to the thermal noise calculation.

**Figure 5 sensors-22-03450-f005:**
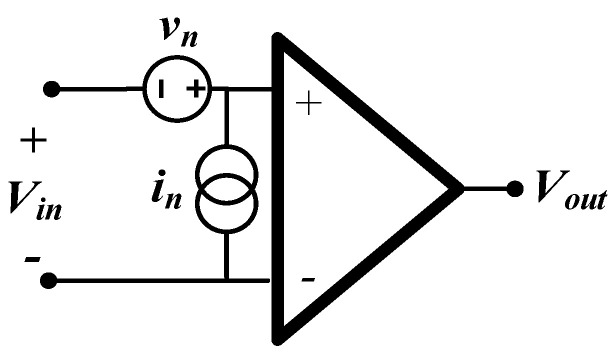
Noise equivalent circuit of the amplifiers. The input impedance of the amplifiers is assumed to be infinite.

**Figure 6 sensors-22-03450-f006:**
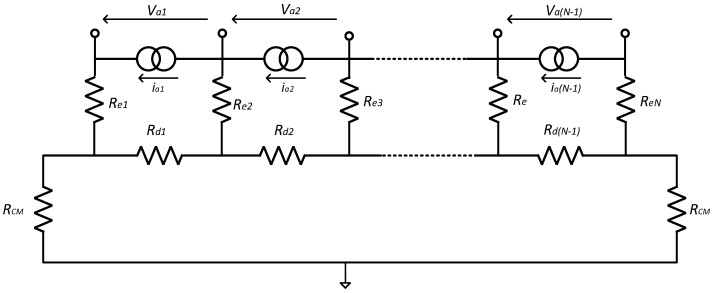
Simplified passband equivalent circuit for amplifier current noise analysis applicable to both Type 1 and Type 2 circuits.

**Figure 7 sensors-22-03450-f007:**
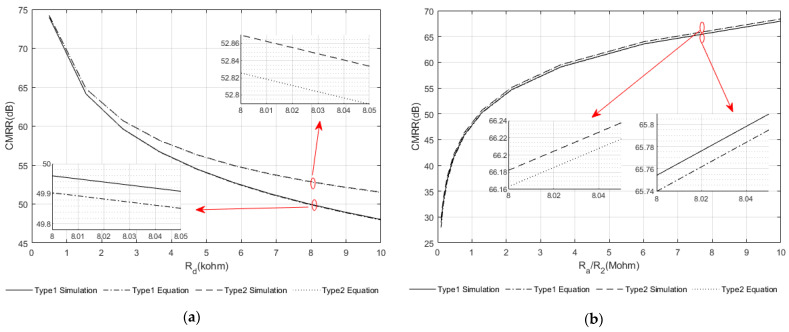
Examples comparing the approximate equations and simulation results for the ‘Type 1’ and ‘Type 2’ interface networks. Minimum CMRR versus (**a**) *R_d_* and (**b**) R_a_ (for Type 1)/*R*_2_ (for Type 2). The nominal component values are: *R*_2_ = *R_a_* = 10 MΩ, *R*_1_ = 10 kΩ, *R_cm_* = *R_d_* = 1 kΩ. Note that the comparisons between calculation and simulation for the other parameters (i.e., crosstalk, thermal noise and amplifier current noise are not plotted, but the limiting values are quoted in the text).

**Figure 8 sensors-22-03450-f008:**
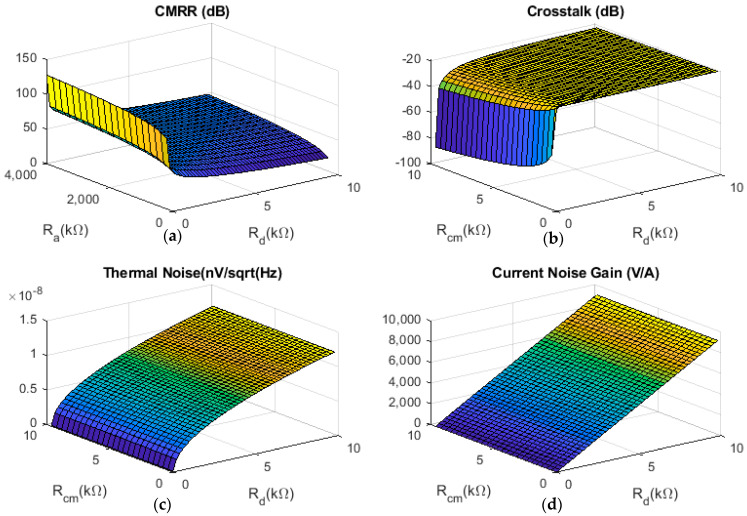
The effect of design parameters on (**a**) CMRR; (**b**) crosstalk; (**c**) thermal noise and (**d**) current noise gain. This 3D presentation of the material from Section 2 and Section 4 is discussed in Section 5.

**Table 1 sensors-22-03450-t001:** Connection matrix **A** for ‘Type 1’ front-end network.

	1	2	…	10	11	12	13	14	20
1	*G_e_*_1_ *+ G_a_*_1_	0	*…*	0	*−G_e_* _1_	0	0	0	0
2	0	*G_e_*_2_ *+ G_a_*_2_	*…*	0	0	*−G_e_* _2_	0	0	0
:	:	:	:	:	:	:	:	:	:
10	0	0	*…*	*G_e_*_10_ *+ G_a_*_10_	0	0	0	0	*−G_e_* _10_
11	*−G_e_* _1_	0	*…*	0	*G_cm_*_1_ *+ G_e_*_1_ *+ G_d_*_1_	*−G_d_* _1_	0	0	0
12	0	*−G_e_* _2_	*…*	0	*−G_d_* _1_	*G_e_*_2_ *+ G_d_*_1_ *+ G_d_*_2_	*−G_d_* _2_	0	0
13	0	0	*…*	0	0	*−G_d_* _2_	*G_e_*_3_ *+ G_d_*_2_ *+ G_d_*_3_	*−G_d_* _3_	0
:	:	:	:	:	:	:	:	:	:
20	0	0	*…*	*−G_e_* _10_	0	0	0	0	*G_cm_*_2_ *+ G_e_*_10_ *+ G_d_*_9_

**Table 2 sensors-22-03450-t002:** Summary of the equations derived in this section (all *R_e_*, *R_a_*, *R_d_* are equal).

Description	‘Type 1’ Circuit	‘Type 2’ Circuit	Equations No.
Common-Mode Gain	$\frac{V_{ai}}{V_{cm}}\approx \frac{\left(5-i\right)R_{d}}{R_{a}}$	$\frac{V_{ai}}{V_{cm}}\approx \frac{\left(5-i\right)(R_{d}||2R_{1})}{R_{2}}$	(5), (6)
Crosstalk Between Channels	$\frac{V_{aj}}{V_{di}}=\{\begin{array}{cc} 1-\frac{R_{d}}{2R_{cm}+9R_{d}} & if\;i=j\\ \frac{-R_{d}}{2R_{cm}+9R_{d}} & if\;i\neq j \end{array}$	$\frac{V_{aj}}{V_{di}}=\{\begin{array}{cc} \frac{R_{d}||2R_{1}}{R_{d}}\left(1-\frac{R_{d}||2R_{1}}{2R_{cm}+9(R_{d}||2R_{1})}\right) & if\;i=j\\ -\frac{R_{d}||2R_{1}}{R_{d}}\left(\frac{R_{d}||2R_{1}}{2R_{cm}+9(R_{d}||2R_{1})}\right) & if\;i\neq j \end{array}$	(11), (12)
Total Thermal Noise Density	$V_{a1}\left(rms\right)=\sqrt{4kT\left(R_{d}+2re(R_{e}\right)}$		(13)
Total Input-Referred rms Noise Density	$v_{a\left(\mathrm{total}\right)}=\sqrt{4kT\left[R_{d}+2re\left(R_{e}\right)\right]+v_{n}^{2}+i_{a}{}^{2}({\left|R_{d}+2R_{e}\right|}^{2}+p{\left|R_{e}\right|}^{2})}\;$		(14)

**Table 3 sensors-22-03450-t003:** Two-wire impedance measurements of the electrodes as dipoles for an implanted cuff in *sheep*.

Electrodes	2—Wire Impedance Measurements (100 mV, 1 kHz)
1–2	2.4 kΩ/−59°
2–3	2.0 kΩ/−58°
3–4	2.6 kΩ/−59°
4–5	3.3 kΩ/−60°
5–6	3.9 kΩ/−51°
6–7	2.5 kΩ/−47°
7–8	1.7 kΩ/−61°
8–9	1.4 kΩ/−59°
9–10	1.3 kΩ/−60°
Reference-1	1.1 kΩ/−48°
Reference-10	1.1 kΩ/−48°

**Table 4 sensors-22-03450-t004:** Common-mode and differential gain for mismatched common-mode impedances (only *R_cm_*s not equal).

Structure	Common-Mode Gain	Differential Mode Gain
‘Type 1’	$\frac{V_{ai}}{V_{cm}}=\frac{\left(5-i\right)R_{d}}{R_{a}}+\left(\frac{2.5R_{d}}{R_{a}\left(1+4.5\frac{R_{d}}{R_{cm}}\right)}\right)\left(\frac{\Delta R_{cm}}{R_{cm}}\right)$	$\frac{V_{ai}}{V_{dj}}=\{\begin{array}{cc} 1-\frac{R_{d}}{2R_{cm}+10R_{d}} & if\;i=j\\ \frac{-R_{d}}{2R_{cm}+10R_{d}} & if\;i\neq j \end{array}$
‘Type 2’	$\frac{V_{ai}}{V_{cm}}=\frac{\left(5-i\right)R_{d}}{R_{2}}+\frac{R_{d}||2R_{1}}{\left(\frac{4R_{2}}{9}\right)\left(1+4.5\frac{R_{d}||2R_{1}}{R_{cm}}\right)}\left(\frac{\Delta R_{cm}}{R_{cm}}\right)$	$\frac{V_{aj}}{V_{di}}=\{\begin{array}{cc} \frac{R_{d}||2R_{1}}{R_{d}}\left(1-\frac{R_{d}||2R_{1}}{2R_{cm}+9(R_{d}||2R_{1})}\right) & if\;i=j\\ -\frac{R_{d}||2R_{1}}{R_{d}}\left(\frac{R_{d}||2R_{1}}{2R_{cm}+9(R_{d}||2R_{1})}\right) & if\;i\neq j \end{array}$

**Table 5 sensors-22-03450-t005:** Common-mode and differential gain for mismatched electrode impedances (*R_d_*s are not equal but *R_a_*s, *R_cm_*s are equal).

Structure	Common-Mode Gain	Differential Mode Gain
‘Type 1’	$\frac{V_{ai}}{V_{cm}}=\frac{\left(5-i\right)R_{di}}{R_{a}}$	$\frac{V_{ai}}{V_{dj}}=\{\begin{array}{cc} 1-\frac{R_{di}}{2R_{cm}+\sum_{x=1}^{10}R_{dx}} & if\;i=j\\ \frac{-R_{di}}{2R_{cm}+\sum_{x=1}^{10}R_{dx}} & if\;i\neq j \end{array}$
‘Type 2’	$\frac{V_{ai}}{V_{cm}}=\frac{\left(5-i\right)R_{di}}{R_{2}}$	

**Table 6 sensors-22-03450-t006:** Variables chosen for validation.

Parameter	Variables Held Constant	Swept Variables
		*both circuits*
CMRR	*R*_2_ = *R_a_* = 10 MΩ	*R_d_*, *R_a_* (*or R*_2_ *for Type 2*)
Crosstalk	*R*_1_ = 10 kΩ	*R_d_*, *R_cm_*
Thermal noise	*R_cm_* = *R_d_* = 1 kΩ	*R_d_*
Amplifier current noise		*R_d_*, *R_cm_*

**Table 7 sensors-22-03450-t007:** Summary of amplifier specifications [14].

Parameter	Specifications
Technology	0.35 µm 4-metal 2-poly CMOS
Power supply	±1.5 V
Midband Gain	79.7 dB
−3 dB frequencies	
Lower	258 Hz
Upper	24.1 kHz
CMRR (@3 KHz)	77.5 dB (A_V,CM_ = 1.29)
PSRR(@3KHz)	
V_DD_	50.57 dB
V_SS_	40.2 dB
Adjacent channel interference(crosstalk)	<−100.1 dB
Total input-referred voltage noise density@3 kHz	$7.5\;\mathrm{nV}/\sqrt{\mathrm{Hz}}$
Total input-referred current noise density@3 kHz	$0.55\;\mathrm{pA}/\sqrt{\mathrm{Hz}}$
Total input-referred rms voltage noise 1 Hz–31 kHz for a source resistance of 1 kΩ	1.82 uV

**Table 8 sensors-22-03450-t008:** Summary of a complete system with Type 1 interface (*R_a_* = 10 MΩ, *R_cm_* = *R_d_* = 1 kΩ, *R_e_* = 0).

Parameter	Calculation	Simulation
Min(CMRR) (@3 KHz)	65.29 dB	65.08 dB
Adjacent channel interference (crosstalk)	−20.83 dB	−20.81 dB
Total input-referred voltage noise density@3 kHz	$9.35\;\mathrm{nV}/\sqrt{\mathrm{Hz}}$	$9.32\;\mathrm{nV}/\sqrt{\mathrm{Hz}}$

## Data Availability

Not applicable.

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
