# Peer review of "The Design of a Low Noise, Multi-Channel Recording System for Use in Implanted Peripheral Nerve Interfaces"

_sensors, 2022, doi:10.3390/s22093450_

Round 1

Reviewer 1 Report

In this paper, the authors discuss the design of a low noise, multi-channel recording system for use in implanted peripheral nerve interfaces. In general, the authors just present their work. The paper lacks of comparison results. Therefore, the conclusions cannot be comprehensively validated. At this point, the authors should further enhance their work.

In section 2.3, the authors discuss the noise analysis. The thermal noise, amplifier voltage noise and amplifier current noise are discussed. Besides, these noises are considered to be Gaussian model. The reviewer wanders to know whether the ambient noise should be considered. Besides, the noise subjected to non-Gaussian model is currently considered in many circumstances such as both papers listed below this comment. At this point, the authors should discuss this issue in their paper. With this operation, the noise analysis would be much more comprehensive.

X Zhang,et al.Parameter estimation of underwater impulsive noise with the Class B model.IET Radar, Sonar & Navigation,2020,Doi: 10.1049/iet-rsn.2019.0477.

Mahmood, et al, “Modeling Colored Impulsive Noise by Markov Chains and Alpha-Stable Processes,” in OCEANS 2015 MTS/IEEE, (Genoa, Italy), May 2015.

The authors design many circuits in their paper. However, the reviewer wanders to know what the difference between their circuits and traditional circuits. The authors should highlight their difference compared to traditional ones.

The testing of authors’ circuits should be presented in this paper. With this operation, the readers can easily understand the authors’ work.

In section 4, the authors just validate their work with their method. However, the readers cannot find the comparison results with traditional methods. At this point, the authors should compare their work to traditional work.

Reviewer 2 Report

The manuscript is well-written and is informative to the reader
on the nerve cuff based neural interface front-end circuitry.

  1. Would the authors expect a difference in performance if the electrode size and separation changed (i.e increased or decreased)? I am assuming the analysis present here is for 1mm diameter, a pitch of 1.5 mm (Ln 379).
  2. How would changes in tissue impedance (i.e under chronic 
    conditions affect the system where it could change by an order due to corrosion or biological changes? Would it become more like single-
    ended channel?
  3.  The text has several minor grammatical/spelling errors (i.e Ln 269, Ln 284). Towards the end of the manuscript, the Table numbers are very mismatched in the table matrix and in text.

Round 2

Reviewer 1 Report

The references are not marked by [1], [2]…., and the references are not listed completely. The authors should carefully proofread the paper before acceptance.